# Association of N-Acetyl Asparagine with QTc in Diabetes: A Metabolomics Study

**DOI:** 10.3390/biomedicines10081955

**Published:** 2022-08-12

**Authors:** Giacomo Gravina, Melissa Y. Y. Moey, Edi Prifti, Farid Ichou, Olivier Bourron, Elise Balse, Fabio Badillini, Christian Funck-Brentano, Joe-Elie Salem

**Affiliations:** 1Institute of Neuroscience and Physiology, Sahlgrenska Academy, University of Gothenburg, 41390 Gothenburg, Sweden; 2Department of Pharmacology and Clinical Investigation Centre (CIC-1901), Pitié-Salpêtrière Hospital, AP-HP, Sorbonne Université, INSERM, F-75013 Paris, France; 3Department of Cardiovascular Sciences, East Carolina University (ECU) Health Medical Center, East Carolina University, Greenville, NC 27834, USA; 4IRD, Unité de Modélisation Mathématique et Informatique des Systèmes Complexes, Sorbonne Université, UMMISCO, F-93143 Bondy, France; 5Nutrition et Obesities, Systemic Approaches, NutriOmique, Hôpital Pitié-Salpêtrière, AP-HP, Sorbonne Université, INSERM, F-75013 Paris, France; 6ICAN Omics, Foundation for Innovation in Cardiometabolism and Nutrition (ICAN), Hôpital Pitié-Salpêtrière, F-75013 Paris, France; 7Sorbonne Université Médecine, Assistance Publique Hôpitaux de Paris (APHP), Service de Diabétologie, Hôpital Pitié-Salpêtrière, INSERM UMRS_1138, Centre de Recherche des Cordeliers, Institute of Cardiometabolisme and Nutrition (ICAN), F-75013 Paris, France; 8Institute of Cardiometabolism and Nutrition (ICAN), INSERM, Sorbonne Université, UMR_S1166, F-75013 Paris, France; 9AMPS LLC, New York, NY 10041, USA

**Keywords:** metabolomics, diabetes, N-Acetyl asparagine, QTc

## Abstract

Changes in the cardio-metabolomics profile and hormonal status have been associated with long QT syndrome, sudden cardiac death and increased mortality. The mechanisms underlying QTc duration are not fully understood. Therefore, an identification of novel markers that complement the diagnosis in these patients is needed. In the present study, we performed untargeted metabolomics on the sera of diabetic patients at a high risk of cardiovascular disease, followed up for 2.55 [2.34–2.88] years (*NCT02431234*), with the aim of identifying the metabolomic changes associated with QTc. We used independent weighted gene correlation network analysis (WGCNA) to explore the association between metabolites clusters and QTc at T1 (baseline) and T2 (follow up). The overlap of the highly correlated modules at T1 and T2 identified N-Acetyl asparagine as the only metabolite in common, which was involved with the urea cycle and metabolism of arginine, proline, glutamate, aspartate and asparagine. This analysis was confirmed by applying mixed models, further highlighting its association with QTc. In the current study, we were able to identify a metabolite associated with QTc in diabetic patients at two chronological time points, suggesting a previously unrecognized potential role of N-Acetyl asparagine in diabetic patients suffering from long QTc.

## 1. Introduction

Cardio-metabolomic and hormonal conditions including diabetic patients have been associated with increased cardiovascular (CV) mortality from cardiomyopathies, ischemic events and sudden cardiac death (SCD) [1]. The pathophysiology of SCD in diabetic patients has been shown to be multifactorial secondary to hypo- and hyperglycemia with associated disorders of potassium abnormalities, autonomic neuropathy and inflammatory and fibrotic changes of the ventricular myocardium [2]. Malignant ventricular arrhythmias (VAs) stemming from a prolonged QT interval occur more frequently in diabetic patients compared to the general population [3]. QT is the time interval between QRS start and T-wave end, reflecting overall cardiac repolarization time [4,5]. When corrected for heart rate (QTc), a prolonged QTc above 480 ms (and even more if >500 ms) is a risk marker for a peculiar form of life-threatening polymorphic VA, called torsade de pointes [6,7]. Determinants of QTc are not fully elucidated and the identification of new markers and pathways is needed to better delineate the pathophysiology of QT dynamics in the diabetic population.

Metabolomics, which involves the study of metabolomes (characteristic set of metabolites or low-molecular-weight components) from a variety of biological samples in a particular condition [8], is an emerging field that can complement clinical diagnostics, identify prognostic indicators to reveal new potential mechanisms associated with specific diseases and assess targeted therapies. In CV disease (CVD), metabolomics has been applied to identify risk factors and understand molecular mechanisms in cardiomyopathies and coronary artery disease (CAD) [9]. The use of metabolomics to assess the risk of QTc prolongation is currently limited to only two studies in the published literature, which assessed the metabolic profiles in an animal model exposed to a fluroquinolone [10] and in a small study of patients who were shift workers [11]. 

To the best of our knowledge, there are no metabolomic studies that assessed the metabolic profile associated with prolonged QTc in diabetic patients. We used an untargeted metabolomics analysis on the sera of study participants with type 2 diabetes mellitus (T2DM) [12], to identify the metabolomic changes associated with QTc. Employing metabolomics analyses, we were able to identify N-Acetyl asparagine as a common molecule in both time points.

## 2. Materials and Methods

### 2.1. Study Design

Our study is a sub-analysis of the *Diabète et Calcification Arterielle* (DIACART) study, a single-center prospective observational cohort study among French patients with T2DM (ClinicalTrials.gov: NCT02431234). The participants were prospectively enrolled at Pitié-Salpêtrière Hospital (Paris, France) and included type 2 diabetic patients (T2DM), at high risk for CVD [13]. A total of 170 patients at baseline visit (T1) and 139 patients at T2 (follow-up for 2.55 years [IQR 2.34–2.88 years]) were included in the study. Patients were excluded if they had severe chronic kidney disease or end-stage renal failure (estimated glomerular filtration rate < 30 mL/min using the Modification of Diet in Renal Disease equation). As QTc becomes unreliable in conditions with prolonged QRS (>130 ms), multiple premature ventricular contractions, ventricular pacing and supraventricular tachycardia, these conditions were excluded from this study [13]. All patients provided written consent and the study was approved by the institutional ethics committee. The data concerning the progression of their peripheral limb arterial disease have recently been published elsewhere [14].

### 2.2. Metabolomics Study

#### 2.2.1. Reference Compounds and Reagents

All liquid chromatography mass spectrometry (LC-MS) grade reference solvents, acetonitrile (ACN) and methanol (MeOH) were from VWR International (Plainview, NY, USA). LC grade ammonium formate and formic acid were from Sigma-Aldrich (Saint Quentin Fallavier, France). Stock solutions of stable isotope-labeled mix (Algal amino acid mixture-13C, 15N) for the metabolomic approach were purchased from Sigma-Aldrich (Saint Quentin Fallavier, France).

#### 2.2.2. Sera Preparation for Metabolomic Analyses

Eight volumes of frozen (−20 °C) acetonitrile containing the internal standards (12.5 µg/mL labeled amino acid mixture) were added to 100 µL of serum samples and vortexed. The samples were sonicated for 15 min and centrifuged for 2 min at 10,000× *g* at 4 °C. The centrifuged samples were then incubated at 4 °C for 1 h to precipitate the proteins. The samples were centrifuged at 20,000× *g* at 4 °C and the supernatants were transferred to another set of tubes to be dried-up and frozen at −80 °C. Samples were reconstituted with the starting mobile phase composition of the chromatographic column and transferred to vials prior to LC-MS analyses.

#### 2.2.3. Ultra-Performance Liquid Chromatography-Mass Spectrometry (UPLC-MS) Analyses of Serum Samples

Metabolomic preparation of samples for analyses was detailed previously [15]. In brief, LC-MS experiments were performed using PFPP, Discovery HS F5-PFPP column, 5 µm, 2.1 × 150 mm (Sigma, Saint Quentin Fallavier, France) on a UPLC^®^ Waters Acquity (Waters Corp, Saint-Quentin-en-Yvelines, France) and Q-Exactive mass spectrometer (Thermo Scientific, San Jose, CA, USA). LC-MS raw data were first converted into mzXML format using MSconvert tool [16]. Peak detection, correction, alignment and integration were processed using the XCMS R package with CentWave algorithm [17] and workflow4metabolomics platforms [18]. The resulting dataset was Log−10 normalized, filtered and cleaned based on quality control (QC) samples [19]. Features were annotated based on their mass over charge ratio (m/z) and retention time using a local database including commercial standards as described previously [20]. The remaining unknown features were discarded from the dataset.

### 2.3. Weighted Gene Co-Expression Network Analysis and Visualization

Standard WGCNA procedure was followed to create unsigned gene co-expression networks from the WGCNA R-package v1.68 [21]. Gene cluster dendrogram was constructed with a power value = 3. A total of 132 metabolites were imported for the WGCNA analysis. The modules identified by WGCNA analysis were further associated, using Spearman correlation, with QTc independently at T1 and T2. The results identified at T1 and T2 were overlapped using Venn diagrams in R software and visualized using MetScape [22] in Cytoscape software [23].

### 2.4. Electrocardiography Acquisition and QTc Analysis

Electrocardiograms were recorded using a digital electrocardiograph by trained personnel, with a sampling rate of 1000 Hz and a filter of 150 Hz. The methodology for QTc assessment (CalECG software^®^, New York City, NY, USA) on 10 s triplicated digitized ECG, Bazett’s heart rate correction) has been extensively detailed elsewhere. Inter- and intra-observer variability assessment for QTc measurement was also performed as has been previously described [4].

### 2.5. Statistical Analysis

Statistical analyses were performed in R-software (version 1.4.1106, RStudio) using “nlme” package, random effect = subjects; fixed effects: age, sex and N-Acetyl asparagine). Correlation with *p*-values < 0.05 were considered significant.

## 3. Results

### 3.1. Baseline Demographics and Clinical Characteristics 

Demographic and clinical characteristics of diabetic patients included in this study are shown in Table 1 and were published in a previous study [13]. The mean age and QTc were 63.9 ± 8.4 years and 422 ± 24.9 ms, respectively, at the baseline visit (T1) and 132/170 subjects were male (77.7%). Mean age and QTc at follow-up visit (T2) were 66.8 ± 8.4 years and 424.9 ± 24.3 ms, respectively, and 112/139 subjects were male (80.6%). A total of 8/139 (5.8%) patients developed an acute coronary syndrome during follow up.

### 3.2. Untargeted Metabolomics Analysis Using Weighted Gene Correlation Network Analysis (WGCNA)

To capture the full extent of the metabolomics expression profiles, we performed a weighted gene co-expression network analysis (WGCNA). WGCNA is a useful method to tightly link co-expressed gene modules to phenotypic traits (e.g., QTc) [21]. The full dataset of 170 patients at T1 and 139 patients at T2 was used for WGCNA analysis. No outliers were identified; therefore, all samples were included in the study analyses. At T1, there were a total of ten clusters of metabolites identified (Figure 1A, left panel). Correlations between clusters of metabolites and QTc were evaluated through Spearman’s correlation analysis. QTc was significantly correlated with the dark grey and the pink clusters (r = 0.25, *p*-value = 0.0008 and r = −0.21, *p*= 0.005, respectively) (Figure 1B, left panel). Similarly, at T2, seven clusters of metabolites were identified (Figure 1A, right panel), among which only the red cluster was correlated with QTc (r = −0.26 and *p* = 0.002) (Figure 1B, right panel). 

### 3.3. Identification of N-Acetyl Asparagine 

These clusters identified by the WGCNA analysis were then compared to identify the common metabolites that correlate with QTc at both time-points (Figure 2A). The intersection of the significant clusters identified N-Acetyl asparagine as the only metabolite in common, pointing to a potential significant involvement of this amino acid in QTc dynamics (Figure 2A). Using the MetScape Cytoscape plugin, we identified the N-Acetyl asparagine enzymatic pathway (including its precursor L-asparagine), as illustrated in Figure 2B. Other pathways associated with N-Acetyl asparagine include the urea cycle and metabolism of arginine, proline, glutamate, aspartate and asparagine.

### 3.4. N-Acetyl Asparagine Positively Correlates with QTc

To evaluate the potential involvement of N-Acetyl asparagine with QTc, we performed Spearman correlation analysis to evaluate the actual relationship between QTc and N-Acetyl asparagine. We found that in both T1 and T2, N-Acetyl asparagine was positively correlated with QTc (r = 0.19, *p* = 0.01 and r = 0.23, *p* = 0.007, respectively) (Figure 3A). In contrast, L-asparagine was negatively correlated with QTc at T2 (r = −0.18, *p* = 0.03) (Figure 3A). We further confirmed the association of N-Acetyl asparagine with QTc in this cohort by applying mixed models (R-software, package ‘nlme’, random effect = subjects; fixed effects: sex and N-Acetyl asparagine) integrating the evolution over time of N-Acetyl asparagine circulating levels in the same subject (Figure 3B). 

## 4. Discussion

To the best of our knowledge, our study was the first to explore metabolomic pathways as potential determinants of QTc in diabetic patients, demonstrating a significant association between N-Acetyl asparagine and QTc.

Metabolomics is a powerful approach at the forefront of scientific discoveries in CVD with its role emphasized in a scientific statement by the American Heart Association [25]. Metabolomics has the potential to reflect the molecular processes more proximal to a disease state by measuring the final downstream products of biological pathways and thereby crucial for the identification of novel biomarkers for risk prognostication and specific therapeutic targets in cardiomyopathies, dyslipidemia treatments and CAD [9,26]. 

In comparison to CAD, heart failure and diabetes, for which metabolomics has been frequently applied to understand the pathophysiology and prognostic risk factors, there are limited metabolomic studies in QTc prolongation and the risk of SCD. On the pathophysiological basis of ischemic changes resulting in metabolic derangements in the cardiomyocyte with decreased ATP and increased reactive oxygen species, Wang and colleagues used proton nuclear magnetic resonance (H^1^-NMR-based) and gas chromatography–mass spectrometry (GC-MS) myocardial tissue metabolic profiling to identify the metabolic alterations in a rat animal model that developed SCD after myocardial infarction [27]. Among 34 rats, 13 developed lethal ventricular arrhythmias (VA, ventricular tachycardia/fibrillation) and 7 developed atrioventricular blocks (AVB) resulting in SCD. There were higher levels of isoleucine, lactate, glutamate, choline, phosphorylcholine, taurine and asparagine, and decreased levels of alanine, urea, phenylalanine, linoleic acid, elaidic acid and stearic acid associated with VA-related SCD. In comparison, only glutamate was elevated and urea was lowered in AVB-related SCD [27]. These metabolic changes were felt to be a result of the detrimental effects of ischemia leading to a depletion in myocardial energy stores and the dysfunction of myocardial membranes altering the cardiac ion channels. 

In the current study, using WGCNA analysis, we identified a significant association between N-acetyl asparagine with QTc in diabetic patients at two distinct chronological time points, which was further confirmed using multivariate mixed models. Using a similar metabolomics approach, in two cohorts of CAD patients, Mehta et al. identified the asparagine pathway as 1 of 7 metabolic pathways in two cohorts of CAD patients. It was an independent predictor of mortality, which was further translated into a prognostic biomarker for all-cause death [28]. Among the different metabolites previously identified in myocardial energetics and inflammation, the aspartate/asparagine pathway exerts a crucial role as anaplerotic substrates in the Krebs cycle during anoxia [29]. Asparagine is converted to aspartate that is then transaminated to oxaloacetate, a critical early metabolite in gluconeogenesis, and plays a key role in the regulation of Krebs cycle intermediates [30]. The appropriate regulation of these metabolites is especially important during ischemia, in which imbalance in the aspartate/asparagine pathway and the inhibition of aspartate has been shown to increase the susceptibility to ischemic damage [31].

In a non-diabetic study cohort, a metabolomics study which assessed metabolomic patterns associated with QTc in small population of 32 male shift-workers from Italy demonstrated a positive correlation between QTc, obesity and hyperglycemia [11]. There were also higher lactate and glucose metabolite levels associated with prolonged QTc while lysine, pyroglutamate, 3-hydroxybutyrate, acetate and glutamine were inversely correlated. Their clinical and metabolomic findings associated with prolonged QTc were suggestive of a metabolic imbalance shift towards anaerobic metabolism or ketosis that can be observed with metabolic syndrome, diabetes and insulin resistance. 

The identification of early metabolic markers and strategies to prevent the development of CV complications (heart failure, CAD and SCD) in diabetic patients is of particular importance, as patients may often not manifest any symptoms of these CV comorbidities. In a large cohort study of 14,294 deaths in Denmark, SCD was found to be 8–9 times higher among young diabetic patients aged 1–35 years old and approximately 6 times higher among those aged 36–49 years old with diabetes than non-diabetic patients [1]. Importantly, within the younger group of diabetic patients, the common cause of SCD was due to ‘sudden arrhythmic death syndrome’ while CAD or ischemia was more common in the older patient population [1]. Previous studies have demonstrated hypo- and hyperglycemia [32], autonomic neuropathy [33], electrolyte abnormalities and ventricular inflammation [2] as contributors to alterations in the cardiac ion channels and increased risk for SCD in the diabetic population. These biochemical and physiologic changes are typically not manifested as symptoms in the disease process and thus the use of metabolomics would be crucial for early recognition of the risk of developing prolonged QTc and subsequent SCD. An upcoming and highly anticipated study, the Recognition of Sudden Cardiac Arrest Vulnerability in Diabetes (RESCUED) project, plans to assess the metabolomic profiles of Dutch patients with T2DM and SCD from the Amsterdam Resuscitation Studies (ARREST) Registry, the Hoorn Diabetes Care System (DCS) and local family practitioner electronic health records to further elucidate the clinical and metabolic factors to prognosticate the risk of SCD in these patients [34]. 

There are limitations to our study which first include that this was a single-center, observational study and thus, a causal relationship cannot be established. While our study identified a particular metabolomic pathway associated with prolonged QTc, future studies involving in vitro experiments such as with cardiomyocytes derived from pluripotent stem cells [5] as well as in vivo interventional studies may be useful to further assess the underlying mechanisms of these findings. Additionally, relatively few subjects had abnormal QTc values in our cohort; which may have limited the discovery of potential associations between other metabolites beyond N-acetyl asparagine and QTc. As our study cohort was limited to a single center of patients in France and with diabetes, the relevance of our findings beyond the geographical location and diabetic population needs to be further evaluated.

## 5. Conclusions

In conclusion, our study employed an untargeted metabolomics approach and identified an association between N-acetyl asparagine with prolonged QTc in diabetic patients. Our findings further highlight the potential role of asparagine metabolism in cardio-metabolic disease and deserve further validation and investigation. 

## Figures and Tables

**Figure 1 biomedicines-10-01955-f001:**
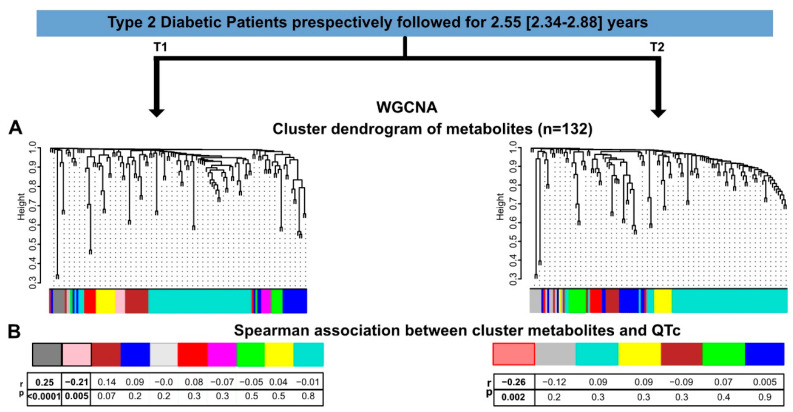
**Untargeted metabolomics analysis using weight gene correlation network analysis (WGCNA)**. Type 2 diabetic patients from the DIACART study were prospectively enrolled and followed up over 2.55 [IQR 2.34–2.88] years. There were 10 clusters of metabolites identified at T1 (**A**, **left panel**) with the dark grey and pink clusters significantly correlated with QTc (**B**, **left panel**). At T2, there were 7 clusters of metabolites identified (**A**, **right panel**) with the light red cluster correlated with QTc (**B**, **right panel**).

**Figure 2 biomedicines-10-01955-f002:**
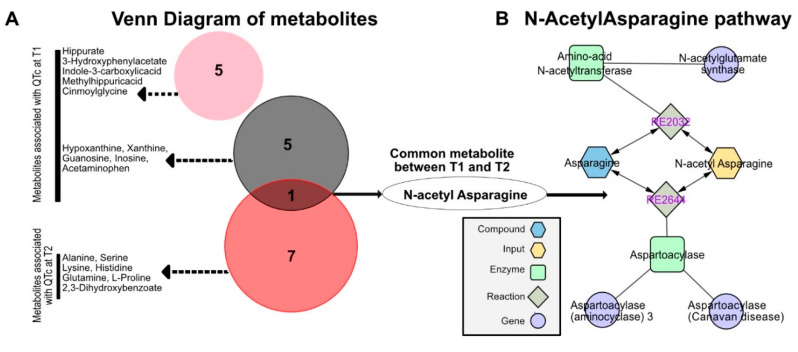
**Identification of individual metabolites at T1 and T2.** (**A**) A total of 10 metabolites were identified at T1 and 7 metabolites were identified at T2 associated with QTc. The intersection of the significant clusters identified N-Acetyl asparagine as the only metabolite in common. (**B**) N-Acetyl asparagine enzymatic pathway.

**Figure 3 biomedicines-10-01955-f003:**
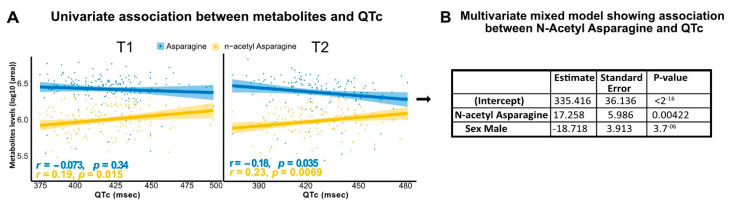
**Association between metabolites and QTc.** (**A**) N-Acetyl asparagine was positively corrected with QTc at both T1 and T2. L-asparagine was negatively correlated with QTc at T2. (**B**) Multivariate mixed model showed a significant correlation with N-Acetyl asparagine and QTc.

**Table 1 biomedicines-10-01955-t001:** Baseline demographics, clinical and electrocardiographic characteristics (adapted from [13]. No statistical differences were identified between T1 and T2, except for age.

	Measurement at T1 (N = 170)	Measurement at T2 (N = 139)
General Characteristics		
Age, years (mean ± SD)	63.9 ± 8.4	66.8 ± 8.4
Male, n (%)	132 (77.7)	112 (80.6)
Weight, kg (mean ± SD)	83.4 ± 15.3	83.2 ± 15.8
Height, m (median [IQR])	1.71 (1.65–1.76)	1.71 (1.65–1.76)
BMI, kg/m^2^ (mean ± SD)	28.9 ± 4.7	28.8 ± 4.9
History of CAD ^a^, n (%)	110 (64.7)	90 (64.7)
Hypertension, n (%)	137 (80.6)	119 (85.7)
**Metabolic Biochemistry Profile**		
HbA1c, % (median [IQR])	7.5 (7.0–8.3)	7.6 (6.9–8.3)
Blood glucose, mmol/L (median (IQR))	7.8 (6.4–9.3)	8.1 (6.6–10.4)
Triglycerides, mmol/L (median (IQR))	1.2 (0.8–2.0)	1.4 (0.9–2.0)
Total cholesterol, mmol/L(median (IQR))	3.7 (3.2–4.4)	3.8 (3.4–4.4)
HDL cholesterol, mmol/L(median (IQR))	1.1 (0.9–1.3)	1.1 (0.9–1.3)
LDL cholesterol, mmol/L (median (IQR))	1.9 (1.5–2.4)	1.9 (1.6–2.4)
**QT Prolonging Drugs** ^b^		
Present, n (%)	8 (4.7)	8 (5.8)
**Basic Metabolic Profile**		
Calcium ^c^, mmol/L (mean ± SD)	2.3 ± 0.1	2.3 ± 0.1
Potassium, mmol/L (mean ± SD)	4.7 ± 0.4	4.6 ± 0.4
Creatinine, mmol/L (median [IQR])	84 (74–101)	87 (76–105)
Albumin, g/L (median [IQR])	42.5 (39.8–44.4)	43 (40.5–45)
**Electrocardiogram Variables**		
Heart rate, beats/min (mean ± SD)	69.7 ± 11.1	69.1 ± 11.8
QTc, ms (mean ± SD)	422 ± 24.9	424.9 ± 24.3

Abbreviations: BMI = body mass index; CAD = coronary artery disease; HbA1c = hemoglobin A1c; HDL = high density lipoprotein; IQR = interquartile range; LDL = low density lipoprotein; QTc = Bazett’s QTc; ^a^ CAD defined as a history of myocardial infarction; coronary angioplasty or bypass grafting; ^b^ Patients taking a drug at known risk of torsades de pointes (www.crediblemeds.org [24]: at T1, these drugs included amiodarone (n = 3), domperidone (n = 1), escitalopram (n = 1) and sotalol (n = 3); at T2, all 8 patients were taking amiodarone. ^c^ Corrected for albumin concentrations.

## Data Availability

The data that support the findings of this study are available from the authors upon request.

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
