# Peer review of "Association of N-Acetyl Asparagine with QTc in Diabetes: A Metabolomics Study"

_biomedicines, 2022, doi:10.3390/biomedicines10081955_

Round 1
Reviewer 1 Report
General comments: The authors reported the association of N-acetyl asparagine with QTc in diabetic patients based on untargeted metabolomic analysis. The experimental design and methodology were appropriate. The manuscript in general was well written. The findings may be helpful for the prognosis of CVD among diabetic patients. I have a few suggestions and inquiries in my specific comments.
Specific comments:
Lines 50-55: The importance of QTc should be better described in the introduction to justify for the current investigation.
Line 96: Change “ml” and “μl” to “mL” and “μL” throughout the manuscript.
Line 134: Did the authors perform p value correction?
Table 1: Why were some of the values presented as mean [IQR]? Were they not normally distributed? If so, reporting median would be more appropriate.
Line 191: Though significant, the correlations seem to be weak.
Figure 3: It seems that relatively few subjects have abnormal QTc values. I wonder if this would limit the discovery of potential associations between metabolites and QTc.
Reviewer 2 Report
The article is very interesting and clear. The results are well describe.The data are preliminary, that also the authors have indicated in the conclusion.
Reviewer 3 Report
Interesting and well written study which needs some minor revision:
1. please include p-values in table 1.
2. please include in a new table the number and type of cardiovascular events occurred during the follow-up period in this cohort of patients with type-2 diabetes.
Round 2
Reviewer 1 Report
The authors have addressed all my comments and revised the manuscript accordingly.
Author Response
thanks for you help